# Studying the Impact of Persistent Organic Pollutants Exposure on Human Health by Proteomic Analysis: A Systematic Review

**DOI:** 10.3390/ijms232214271

**Published:** 2022-11-17

**Authors:** Sophie Guillotin, Nicolas Delcourt

**Affiliations:** 1Poison Control Centre, Toulouse University Hospital, 31059 Toulouse, France; 2INSERM UMR 1295, Centre d’Epidémiologie et de Recherche en Santé des Populations, 31000 Toulouse, France; 3INSERM UMR 1214, Toulouse NeuroImaging Center, 31024 Toulouse, France

**Keywords:** persistent organic pollutant, proteomics, hepatotoxicity, reprotoxicity, developmental toxicity, neurotoxicity, cardiotoxicity, immunotoxicity, endocrine disruptors

## Abstract

Persistent organic pollutants (POPs) are organic chemical substances that are widely distributed in environments around the globe. POPs accumulate in living organisms and are found at high concentrations in the food chain. Humans are thus continuously exposed to these chemical substances, in which they exert hepatic, reproductive, developmental, behavioral, neurologic, endocrine, cardiovascular, and immunologic adverse health effects. However, considerable information is unknown regarding the mechanism by which POPs exert their adverse effects in humans, as well as the molecular and cellular responses involved. Data are notably lacking concerning the consequences of acute and chronic POP exposure on changes in gene expression, protein profile, and metabolic pathways. We conducted a systematic review to provide a synthesis of knowledge of POPs arising from proteomics-based research. The data source used for this review was PubMed. This study was carried out following the PRISMA guidelines. Of the 742 items originally identified, 89 were considered in the review. This review presents a comprehensive overview of the most recent research and available solutions to explore proteomics datasets to identify new features relevant to human health. Future perspectives in proteomics studies are discussed.

## 1. Introduction

Persistent organic pollutants (POPs) cover a set of chemical substances that have four properties that the Stockholm Convention explained in 2001 [1]. They are persistent, bioaccumulative, toxic, and mobile. POPs mainly come from emissions that have been released into the environment for several decades from human activities. The pollutants slowly degrade (taking a few years to centuries); accumulate in living beings over time, particularly in adipose tissues and in the food chain; and are likely to cause harmful effects. POPs are also mobile and high concentrations can be measured far from emission sources; they are found in riverbeds where they were discharged, but also in locations where they have never been used. The pollution of ecosystems, living organisms, and many foodstuffs leads to long-term exposure to harmful chemicals for many species, including humans.

Thus, the human population is exposed daily to chemical pollutants through the environment and food, which accumulate in all living beings, particularly in adipose tissue due to their lipophilic properties. Recent studies suggest that chronic exposure to these environmental contaminants could be the cause of the development of endocrine, metabolic, immunological, and neurological pathologies. These substances are also known to be reprotoxic and promote the occurrence of cancer.

The 2001 Stockholm Convention on Persistent Organic Pollutants aims at eliminating twelve chemicals that are of particular concern for human health because they are highly toxic, bioaccumulative, difficult to degrade, and disseminate over long distances. These POPs are classified in three annexes according to whether their use and production must be eliminated (Annex A), restricted (Annex B), or their uncontrolled emission into the environment must be reduced or even eliminated (Annex C). These twelve persistent products covered by the Stockholm Convention are called the “Dirty Dozen”. They include aldrin, chlordane, DDT (dichlorodiphenyltrichloroethane), dieldrin, endrin, heptachlor, mirex, toxaphene, PCBs (polychlorinated biphenyls), hexachlorobenzene, dioxins, and furans. In 2009, nine new compounds joined the Dirty Dozen. These are chlordecone, lindane, alpha-hexachlorocyclohexane, beta-hexachlorocyclohexane, octabromodiphenyl ether, pentabromodiphenyl ether, perfluorooctane sulfonic acid, its salts, and perfluorooctane sulfonyl fluoride, hexabromobiphenyl, and pentachlorobenzene.

Due to their impact on human health, POPs have been widely studied by the scientific community. These studies focus on monitoring programs to evaluate the exposition and bioaccumulation of these substances in humans, but also on the molecular and cellular mechanisms underlying their toxicities in in vitro and in vivo models, whether cellular or animal models. In this context, using proteomic-based approaches allows global information to be generated on toxicant-induced molecular and cellular perturbations in cells and tissues, which are associated with adverse outcomes.

In this review, we describe the different proteomic approaches available and their contributions, to better characterize the human health consequences of POP exposure.

### Proteomic Overview

Proteomics aims at profiling the complete protein content of a biological sample, including protein modification and interaction [2]. The mainstream proteomic approach is based on mass spectrometry (MS) technology. In shotgun proteomics, the workflow is based on protein extraction from the biological sample, followed by digestion by an endoprotease (commonly trypsin, which cleaves after each lysine and arginine residue), chromatographic separation, and MS analysis. MS determines the mass-to-charge ratio (m/z) of each tryptic peptide in a complex mixture of thousands of peptides and further fragments of each peptide to allow the determination of the amino acid sequence [2]. All these data are then computationally analysed against protein databases that provide the identity of each protein. The latest technological advances in MS have made it possible to identify several thousand proteins present in biological samples or organisms [3].

To bring the necessary quantitative dimension to the characterization of protein changes, several approaches have been described. Among them, 2-dimensional electrophoresis (2DE) prior to MS is a common methodology used to separate proteins by mass and pI (charge) that can differentially detect expressed protein spots using the difference in gel electrophoresis (DIGE) method [4]. This approach was notably used to study proteome changes in human HepG2 cells and SH-SY5Y cells exposed to endosulfan [5,6]. More recently, Pavlikova et al. described dysregulation in the expression of several proteins involved in the stress response, mitochondrial process, and cell maintenance in NES2Y cells exposed to DDT [7]. Shotgun proteomics is another approach of choice, as it leads to the quantification of thousands of proteins without a priori. The quantification method can be based on protein labelling or label-free methods. First, sample preparation can be combined with chemical labelling using isobaric tag techniques, such as isobaric tags for relative and absolute quantification (iTRAQ). This method is based on the covalent labelling of the N-terminus and side chain amine of tryptic peptides with tags of different mass. The samples are then pooled prior to fractionation and MS/MS analysis. The generated data are used to identify labelled peptides and relatively quantify the peptides and their related proteins from the samples they originated from [8]. Using this quantitative method, Cowie et al. [9] described dysregulation in the proteins described to be involved in Parkinson’s and Huntington’s disease after the exposure of zebrafish (*Danio rerio*) to dieldrin. Another renowned strategy was developed by Mann’s group, based on the labelling of proteins in the biological sample using stable isotopes, such as arginine or lysine ^13^C and/or ^15^N [10]. This method was chosen by Zhang et al. [11] to investigate the molecular pathways perturbed by in vitro exposure of PFOS in mouse embryonic stem cells. More recently, TMT labelling, an approach that allows up to tenfold multiplexing and isotope labelling-based quantification at the same time, was also used to study protein regulation in *Danio rerio* exposed to PFOS [12] and in SKBr3 human cells exposed to PFOA [13]. Associated with advances in MS and bioinformatics tools, label-free quantitative proteomic approaches were developed and are now considered to be reliable, efficient methods of studying protein level changes in complex mixtures. These approaches are based on the measurement of either the MS/MS sampling rate (the rate of signal sampling within a definite time) of a particular peptide (generally the highest), or of its MS chromatographic peak area. These values are then directly related to peptide abundance. Notably, these approaches are compatible with high-throughput quantitative analysis, but they have a lack of reproducibility, particularly when samples have a wide dynamic range [14]. In the field of toxicological research, label-free quantitative proteomics has been used to evaluate the response to PCDD in three transgenic mouse lines, each expressing a different rat AhR isoform (rWT, DEL, and INS) providing widely differing resistance to PCDD toxicity, as well as C57BL/6 and DBA/2 mice, which exhibit a tenfold divergence in PCDD sensitivity [15].

## 2. Results and Discussion

Studies on the contribution of proteomics to the human toxicology of POPs conducted up to 2022 were reviewed, and an increasing trend was observed (Figure 1). This indicates that the study of POPs through proteomics has drawn more research attention in recent years.

Articles addressing the contribution of proteomics to the human toxicology of POPs were categorized according to the different types of toxicity they described (Figure 2, Table 1, Table 2, Table 3, Table 4, Table 5, Table 6 and Table 7). We then proceeded in the reverse direction, describing each molecule in the manuscript, according to the type of toxicity.

### 2.1. Brominated Flame Retardants

Brominated flame retardants (BFRs) are widely used in a variety of industrial and consumer products (e.g., automobiles, electronics, furniture, textiles, and plastics) to reduce flammability. Under the Stockholm Convention, the use of several BFRs is restricted or banned: tetrabromodiphenyl ether (BDE-47), pentabromodiphenyl ether (BDE-99), and decabromodiphenyl ether (BDE-209); three polybrominated diphenyl ethers (PBDEs); and hexabromocyclododecane (HBCD). Unfortunately, these compounds can also contaminate the environment, resulting in human exposure. BFRs can act as endocrine disruptors, and it was suggested that PBDEs may have an adverse effect on thyroid hormones, reproductive hormones, semen quality, and neonatal health [90]. Similarly, BFR exposure can interfere with fundamental aspects of neurodevelopment, altering molecular pathways associated with adverse neurocognitive and behavioural outcomes [91]. Here, we focused on proteomic studies and their contribution to improving knowledge about the toxicology of BFRs.

#### 2.1.1. BDE-47

Ji et al. tested the potential Parkinson disease (PD)-related neurotoxic effects of BDE-47 in mice [81] and rats [82]. Results of the proteomic study of PD-related brain regions revealed significant protein changes in pathways involved in oxidative stress and neurotransmitter production [81]. The proteomic study revealed that protein degradation pathways were affected [82]. Western blot analysis confirmed that BDE-47 could inhibit ubiquitination and autophagy processes, resulting in the increased formation of a-synuclein aggregate, an important pathological hallmark of PD. Overall, the studies suggested that the occurrence of BDE-47 in the brain could be a risk for developing PD.

GC1-spg cells were used to verify the spermatogenesis mechanisms of BDE-47 [75]. Proteomics data indicate that BDE47 disrupts spermatogenesis, impairs mitochondrial function, and induces apoptosis of cells probably via the mitochondrial pathway. Other data improve our understanding of the mechanisms responsible for BDE-47-induced male reproductive toxicity [76]. Based on a proteomic analysis coupled with a bioinformatics analysis using ingenuity pathway analysis (IPA) methods, the authors found that BDE-47 mainly affected molecules involved in oxidative stress, cell regulation, and the inflammatory response.

Neurodevelopment was another key focus of proteomic studies. Song et al. sub-cultured neural stem/progenitor cells that were exposed to BDE-47 [51]. After 72 h of exposure, they found that cofilin-1 and vimentin were differentially expressed in all groups. These results are of particular interest as cofilin is a key regulator of the actin dynamic [92] and vimentin, a member of the intermediate filament, is mainly present in the nervous system. In an in vivo study, juvenile female BALB/C mice were exposed for 28 days to fish-based diets spiked with BDE-47 at doses approximating the LOAEL [54]. It was found that BDE-47 elicited changes in neural protein expression profiles. Calcium homeostasis was also affected by BDE-47 exposure. They concluded that BDE-47 breached the blood–brain barrier (BBB) and accumulated in the juvenile brain where it could induce excitotoxic insults by the dysregulation of calcium homeostasis.

#### 2.1.2. BDE-99

Concentration-dependent differences in protein expression was analysed using 2D-DIGE for cultured cortical cells isolated from rat foetuses after 24 h exposure to BDE-99 [80]. Low-dose BDE-99 exposure induced marked effects on cytoskeletal proteins. Interestingly, BDE-99 exposure increased the expression of phosphorylated Gap43, reflecting effects on neurite extension processes.

More recently, a proteomic approach was used to study the early effects of BDE-99 in two distinct regions of the neonatal mouse brain, the striatum and hippocampus [56]. The authors found that the levels of proteins involved in neurodegeneration and neuroplasticity (e.g., Gap-43) were typically altered in the striatum, and proteins involved in metabolism and energy production were altered in the hippocampus. Significantly, many of the identified proteins have been linked to protein kinase C signalling, which is known to play a role in learning and memory processes [93,94]. These specific responses to early exposure to BDE-99 may contribute to persistent neurotoxic effects in adults.

#### 2.1.3. BDE-209

Song et al. sub-cultured neural stem/progenitor cells that were exposed to BDE-209 [51]. As with BDE-47, the authors revealed that cofilin-1 and vimentin were differentially expressed.

#### 2.1.4. HBCD

Screening of proteome changes in zebrafish liver cells was performed to generate hypotheses regarding exposure to sublethal doses of HBCD [18]. Quantitative analyses revealed that distinct HBCD responses were related to a decreased abundance of protein involved in energetic metabolism. There was also evidence that HBCD may target other metabolic pathways. In two studies, the influence of short-term exposure of rats to HBCD was studied by investigation of the liver proteome [19,20]. Proteome analysis of the liver confirmed that HBCD exposure significantly changed the abundance of proteins involved in metabolic processes (gluconeogenesis/glycolysis, amino acid metabolism, and lipid metabolism) and in the oxidative stress response. The largest sex-dependent effect concerned the concentration of proteins involved in lipid metabolism, which may have led to the considerably higher ratio of HBCD accumulated in the white adipose tissue of exposed female rats than males. The results further elucidate the different sensitivities of sex toward HBCD exposure on gene expression.

Rasinger et al. also exposed juvenile female mice for 28 days to HBCD doses approximating LOAEL [54]. HBCD was also found to be accumulated in the juvenile brain, where it could induce excitotoxic effects. These effects were confirmed after exposure to higher doses of the same pollutant [55].

To conclude, proteomic studies on neuronal cells highlighted that BFRs exert similar neurotoxicological outcomes. Notably, these results suggested that BDE-47 could be a potential neurotoxicant involved in the pathophysiology of PD. Regarding neurodevelopmental effects, several dysregulated proteins were found to be in common following BFR exposures: cofilin-1 and vimentin, cytoskeletal proteins, and proteins involved in calcium homeostasis. In contrast, the cellular mechanisms involved in reprotoxicity were heterogenous: BDE-47 exposure led to the dysregulation of proteins involved in mitochondrial function and oxidative stress, whereas BDE-99 exposure modified the expression of proteins involved in cytoskeleton maintenance. Hepatotoxic effects were only analysed by proteomic techniques for HBCD. An overall decrease in the abundance of protein involved in metabolic processes was noted.

### 2.2. Dioxins

Although dioxins can be released from some natural processes, their presence is mainly due to atmospheric emissions of a number of industrial processes. These include waste incineration, thermal and combustion-related activities, chlorine bleaching of paper pulp, and vehicle traffic. Here, the Stockholm Convention recommends reducing unintentional release of polychlorinated dibenzo-p-dioxins (PCDD) and polychlorinated dibenzofurans (PCDF). It is well-established that these compounds are highly toxic, and can cause adverse reproductive and developmental effects, neurodevelopmental impairment, and immune system damage. In addition, the possibility that PCDD/Fs may cause cancer is especially worrying [95]. In this review, we focus on proteomic studies that reveal new aspects of biological pathways responsible for dioxin toxicity.

#### 2.2.1. PCDD

Many studies have assessed the impacts of PCDD on the liver using proteomic technologies. A study performed on the rat hepatoma cell line 5L [39] suggested that altered protein abundances resulted from an adaptive response to PCDD-induced oxidative stress. The adaptive response to oxidative stress was also observed in Sprague-Dawley rats [40] and humans (automobile emission inspectors and waste incineration workers) [48]. These data point to a mechanism by which PCDD may affect cellular homeostasis and survival. Despite the differences in species, concentration, duration, and technique among studies, there were several common findings, including effects on liver metabolism, such as on the blood coagulation pathway; lipid metabolism; and energy metabolism. Oberemm et al., demonstrated changes of protein expression in the livers of male marmosets that were subjected to a single dose of PCDD [42]. In the liver, transferrins, lamin A, and HSP70 were found to be upregulated. To investigate peptide changes in the sera of rats and explore the association of these changes with liver morphology, PCDD was administrated to male rats for 29 weeks [43]. One peptide, fibrinopeptide A, was found to be significantly decreased. Fatty degeneration and necrosis of the liver were both found to be significantly increased after PCDD exposure. Thus, levels of fibrinopeptide A were significantly correlated with fatty degeneration and necrosis of the liver. The results were confirmed after analysis of plasma samples from workers at municipal incinerators [47]. According to the results, exposures to PCDD may induce liver disease, as suggested by the role of several identified dysregulated proteins (e.g., fibronectin and fibrinogen gamma A). Disruption of lipid metabolism was also reported in a study where two rat groups were exposed to PCDD: one group for short-term exposure and the other for long-term exposure (1 month) [38]. Among the identified proteins, apolipoprotein A-IV is of particular interest as it may protect cells from lipid peroxidation and organisms from atherosclerosis induced by PCDD exposure. As Apo-AIV has a protective role in lipid peroxidation and atherosclerosis, these results are consistent with the elevated lipid levels and liver injury described in the incinerator workers [96]. Finally, potential exposure biomarkers of PCDD were investigated in cultured media from HepG2 cells. In this quantitative analysis, fifteen secreted proteins were found to be dysregulated. Bio-informatic analysis revealed that they are involved in oxidative stress, energy metabolism (glucose, lipid, and amino acid), and blood coagulation [49]. Proteins involved in cell regulation were also impacted by PCDD exposure. A comprehensive quantitative proteome analysis was performed on 5L rat hepatoma cells exposed to PCDD for 8 h [41]. The identified proteins included several proteins implicated in cell cycle regulation, growth factor signalling, and control of apoptosis. In the latest study, the effects of PCDD on protein modification, such as glycosylation and phosphorylation, were extensively studied in Chang human liver cells before and after treatment with PCDD [45]. Data confirmed interesting insights on the molecular and biochemical events of PCDD-mediated toxicity in cellular protein folding and turnover, cytoskeletal networking, and vesicular trafficking. Interestingly, proteomic analyses studying the effects of PCDD exposure in relation to AhR binding were also carried out. A quantitative analysis of changes in protein phosphorylation preceding or accompanying transcriptional activation PCDD in 5L rat hepatoma cells was performed [44]. Most of the TCCD-induced phosphorylation changes have not been previously reported: the transcription factor ARNT (the obligate partner for gene activation by the PCDD-bound AhR) exhibited an upregulation of its Ser77 phosphorylation level, a post-translational modification (PTM) known to control the differential binding of ARNT to DNA. Other proteins with altered phosphorylation included, among others, various transcriptional coregulators not previously known to participate in PCDD-induced gene activation. Another proteomic analysis adopted an approach to identify the differential effects of PCDD exposure on liver protein expression in Han/Wistar rats compared with Long-Evans rats [46]. The results revealed, for the first time, a subset of hepatic proteins that were differentially regulated in response to PCDD in a strain-specific manner. These results suggested that AhR may play a role in establishing the major differences in PCDD response between these strains of rats. A recent study evaluated three transgenic mouse lines, each expressing a different rat AhR isoform providing widely differing resistance to PCDD toxicity [15]. Several hepatic proteins showed parallel upward or downward alterations (SNRK, IGTP, and IMPA2), showing consistent strain-dependent changes.

Perturbed expression of reproduction-related pathways was observed in rats. Male rats were administered PCDD [73]. The proteome and variables relating to spermatogenesis were investigated. PCDD disturbed testicular proteome profiles in rats. In particular, the expression of six testicular proteins involved in the oxidative stress pathway were significantly upregulated. Interestingly, the fertility protein SP22, a potential biomarker to diagnose human infertility, was downregulated. In order to elucidate low-dose PCDD-mediated effects on reproductive or endocrine functions, female rats were also administered PCDD [74]. A proteomic analysis of the ovaries showed distinct changes in the levels of several proteins that were related to the stress response. With regard to in vitro studies, Orlowska et al. aimed to identify proteins potentially involved in the mechanism of PCDD action and toxicity in porcine granulosa cells. Functional analysis showed that cytoskeletal proteins formed the largest class of proteins significantly affected by PCDD. They demonstrated that PCDD may affect ovarian follicle fate by the rearrangement of the cytoskeleton and extracellular matrix, as well as the modulation of proteins important for the cellular response to stress [71]. In their subsequent study, the same authors sought to examine PCDD-induced changes in the proteome of AhR silenced porcine granulosa cells [72]. In AhR-silenced porcine granulosa cells, PCDD influenced the abundance of only three proteins: annexin V, protein disulfide isomerase, and ATP synthase subunit beta. The results revealed the ability of PCDD to alter protein abundance in an AhR-independent manner.

Effects on development-related pathways have been commonly reported. Rasinger et al. exposed 28 day-old juvenile female mice to PCDD at doses approximating LOAEL [54]. PCDD was accumulated in the juvenile brain where it could induce excitotoxic effects. The authors confirmed the effects after exposure to higher doses of the same pollutant [55]. In zebrafish, neurological pathways were also impacted. Shankar et al. hypothesized a WFIKKN1 protease inhibitor role in AhR signalling, and showed that *WFIKKN1* gene expression was Ahr-dependent in developing zebrafish exposed to PCDD [64,97]. Functional enrichment demonstrated that the protein WFIKKN1 was involved in skeletal muscle development and played a role in neurological pathways after PCDD exposure. Impact of PCDD on ovarian development was also investigated by studying the yolk of chicken eggs before the start of development [61]. Exposure to PCDD prior to the start of embryonic development resulted in significant changes in expression of proteins involved in blood clotting, oxidative stress, electron transport, and calcium regulation. Impact on eye development was also studied. Indeed, how exposure to PCDD modulates the acceleration of eye opening remains unknown. To reveal the underlying mechanisms of accelerated eye opening, pregnant mice were injected with PCDD, and tissues around the eye of neonatal mice were subject to proteome analysis [62]. Upon PCDD administration, stathmin 1 level was increased. A hypothetical mechanism could be the proliferation of corneal epithelial cells before eye opening, suggesting that stathmin 1 may be involved in accelerated eye opening, probably by stimulating proliferation of corneal epithelial cells. Lastly, PCDD was given to timed-pregnant C57BL/6J mice [63]. The proteomic changes in the palates of the foetal mice were studied. In PCDD group, the incidence of cleft palate was 100%. Peroxiredoxin-1 was robustly up-regulated in the cleft palate group, as well as proteins linked to energy metabolism, cell migration, and apoptosis. Peroxiredoxin-1 protein may be associated with cleft palate in mice induced by PCDD.

PCDD are chemical substances described to exert cardiotoxic effects. A label-free quantitative proteomic approach investigated the disturbance of the cardiac proteome induced by PCDD in the adult zebrafish heart [86]. The proteins identified as altered by PCDD encompass a wide range of biological functions including calcium handling, myocardium cell architecture, energy production and metabolism, mitochondrial homeostasis, and stress response. Collectively, the results indicated that PCDD exposure alters the adult zebrafish heart in a way that could result in cardiac hypertrophy and heart failure.

Proteins involved in immune pathways were dysregulated. A basis for the striking change in human keratinocyte colony morphology due to PCDD exposure has been investigated using shotgun proteomics [85]. Indeed, alteration of the skin barrier leads to a decrease in epidermal immunity [98]. Proteomic analysis has revealed significant decreases in the differentiation markers filaggrin, keratin 1, and keratin 10. The results suggested that reduced levels of differentiation marker proteins contributed to the observed excessive stratification it induces. In another study, two groups of Sprague-Dawley rats were exposed to PCDD; one group received short-term exposure and the other received long-term exposure [83]. The volume-increased proteins were cytokeratins and immunoglobulins. These proteins stimulated the immune system. In addition, an analysis demonstrated changes of protein expression in the thymus of male marmosets that were administrated a single dose of a PCDD [42]. In the thymus, where the pattern of dysregulated proteins could be clearly related to immune responses, proteins involved in oxidative stress and cytoskeleton maintenance were dysregulated. In another immune function organ, a proteome approach was used to investigate the disturbance of osteogenesis evoked by PCDD in an in vitro osteoblast differentiation model of rat mesenchymal stem cells [84]. Proteins showing a significant change in abundance were mostly involved in cytoskeleton organisation and biogenesis, actin filament-based processes, protein transport, and folding. These results suggest that PCDD-induced decreases in the expression of calcium-binding proteins may interfere with osteoblast calcium deposition, which was, in fact, reduced by PCDD.

Finally, a proteomic analysis of the interaction among multiprotein complexes involved in PCDD-mediated toxicity in urinary bladder epithelial RT4 cells was performed [89]. Major differences between the control proteome and exposed cells concerned calcium- and iron-regulated proteins. As nitric oxide (NO) production was significantly enhanced in PCDD-exposed cells, the authors hypothesized that alterations in calcium and iron homeostasis upon exposure to PCDD may be linked through NO.

#### 2.2.2. PCDF

Using a proteomic approach, a study was conducted in order to determine the effects of PCDF on proteins secreted by HepG2 cells [50]. Among the 32 proteins identified by proteomic analysis, the differential expression of protein DJ-1, proteasome activator complex subunit 1, and plasminogen activator inhibitor-3 was further validated in plasma proteins from rats exposed to PCDF. These proteins could be used as potential biomarkers of PCDF exposure.

Taken together, the results demonstrated a wide range of cellular changes after PCDD exposure on different organs. The role of AhR in PCDD-mediated toxicity was shown in studies regarding hepatotoxic, reprotoxic, and developmental effects. Studies that aimed to evaluate hepatotoxic effects also showed a dysregulation of proteins involved in metabolic processes (especially the blood coagulation pathway and lipid metabolism) and oxidative stress. Interestingly, SP22 was suggested as a potential biomarker of the reprotoxic effects of PCDD. Surprisingly, exposure to PCDF was poorly described by proteomic techniques, and only one study on hepatotoxic effects was reported. Finally, a study on cardiotoxic effects indicated that exposure to PCDD could lead to heart disease.

### 2.3. PCBs

PCBs are used in industry as heat exchange fluids, in electric transformers and capacitors, and as additives in paint, carbonless copy paper, and plastics. There are two categories of PCBs, with a total of 209 different types: dioxin-like compounds and non-dioxin-like compounds. These pollutants globally circulate through the atmosphere, the hydrosphere, and the food chain, so they can be found in places where they have never been produced [99]. The Stockholm Convention recommends their elimination. Indeed, due to this large use in industry, many studies have found different toxic effects: hepatotoxicity [100], neurotoxicity [101], and also immunotoxicity [102]. Here, we focus on proteomic studies to complement this knowledge.

PCB exposure has been associated with liver enzyme elevation and suspected steatohepatitis in cohort studies. Comparative proteomics analyses were performed in mice livers from acute exposure [34] and chronic exposure studies [35,36,37]. Experiments based on acute exposure showed that PCBs produced dose-dependent increases in Cyp1a1 and Cyp2b10, two cytochromes induced by Ahr in PCB metabolism [34]. Experiments based on chronic exposure confirmed that PCB exposure differentially regulated the hepatic proteome according to the Ahr activation [36]. Jin et al. performed a study to better understand the role of AhR by using wild type and AhR-/- mice [37]. The PCB126-associated liver proteome was Ahr-dependent. Ahr principally regulated liver metabolism (e.g., lipids, xenobiotics, and organic acids) and bioenergetics, and it also impacted liver endocrine response and function, including the production of steroids, hepatokines, and pheromone-binding proteins. These effects could have been indirectly mediated by interacting transcription factors or microRNAs. Chronic PCB treatment also demonstrated alterations in the function of 42 transcription factors including downregulation of NRF2 and key nuclear receptors previously demonstrated to protect against steatohepatitis (e.g., HNF4α, FXR, PPARα/δ/γ, etc.) [35]. PCBs impair liver functions by reducing its protective responses against nutritional stress to promote diet-induced steatohepatitis.

Proteins relating to endocrine and reproduction disruption were also impacted by PCBs. In a study, MCF-7 cells were exposed to environmentally relevant concentrations of PCBs [67]. Affected proteins included those regulating oxidative stress, such as superoxide dismutase and structural proteins such as actin or tropomyosin, which may explain morphological cell changes. Lasserre et al. confirmed these biological pathways involved in PCB exposure by analysis on subcellular fractions of MCF7 [68]. Concerning the steroidogenic capacity, the objective was to determine the effects of three structurally different PCB congeners, PCB118, PCB 126, and PCB 153 on the proteome of H295R cells [69]. Exposure to PCBs perturbed steroidogenesis and protein expression. Overall, alterations in protein regulation and steroid hormone synthesis suggest that exposure to PCBs disturbs several cellular processes, including protein synthesis, stress response, and apoptosis. These effects on several cellular processes were also observed by Williams et al. [70]. In this study, the authors investigated the effects of PCB treatment compared with oestradiol treatment on the proteome of the mouse mammary gland. Most oestrogen-independent effects influenced several molecular processes including apoptosis, cell adhesion, and proliferation.

There was also evidence that PCBs may target pathways involved in development. One study aimed to determine whether exposure of pregnant sheep to three different mixtures of PCBs affected fetal testis development [57]. Changes in protein regulation affected cellular processes, such as stress response, protein synthesis, and cytoskeleton regulation. The study demonstrated that in utero exposure to PCB mixtures exerted subtle effects on the developing fetal testis proteome but did not significantly disturb testis morphology and testosterone production. Developmental exposure to PCBs has been also associated with cognitive deficits in humans and laboratory animals by mechanisms that remain unknown. Comparison of the brain proteome from juvenile rats [54,55,59,60] or primary cerebellar neurons from rats [58] unexposed and exposed to PCBs was performed. The biological pathway associated with Ca^2+^ homeostasis (transport and signalling) was mainly disrupted. These perturbations suggest that PCBs may alter energy metabolism and intracellular signalling, and may contribute toward a premature aging proteome profile. Other signalling pathways were dysregulated by the PCB exposure, including cytoskeletal process [58,59], trafficking of proteins [58], phosphoinositol signalling pathway [59], and lipid metabolism [54].

There is a growing body of evidence that POPs may increase the risk for cardiovascular disease, but the mechanisms remain unclear. Protein composition of HDL from 17 subjects were analysed [87]. Pathway analysis demonstrated that proteins involved in lipid metabolism were positively associated with increased PCBs, including cholesteryl ester transfer protein, phospholipid transfer protein, and serum amyloid A. Conversely, proteins involved in negative regulation of proteinases, acute phase response, platelet degranulation, and complement activation were negatively associated with PCB exposure. Moreover, PCB153 has been reported to bind to the oestrogen receptor, induce vessel formation, and increase the formation of reactive oxygen species in endothelial cells. As PCB153-induced phenotypic changes are similar to estradiol, a study postulated that PCB153 activates common redox signalling pathways to 17β-oestradiol [88]. Therefore, they investigated the proteome of HMVEC cells exposed to PCB153. Network analysis showed that TGFB1 and c-Myc play central roles. “Cardiovascular System Development and Function” was found to be associated with PCB exposure in network analysis.

Together, proteomic approaches revealed a variety of molecular effects following cell exposure to PCBs. As showed with PCDD, a role for AhR in PCBs-mediated toxicity was showed in studies regarding hepatotoxic effects. Developmental effects studies showed subtle effects for PCBs on the proteome of the fetal testes but not on morphology and testosterone production, whereas endocrine studies demonstrated an impact of PCBs on steroidogenesis. Interestingly, proteomic studies on brain development highlighted premature ageing after exposure. Finally, PCB exposure may increase the risk of cardiovascular disease by affecting lipid metabolism and vessel formation.

### 2.4. Pesticides

Pesticides are a major class of POPs that have a high resistance to natural biodegradation and an enhanced tendency to bioaccumulate. The excessive and unorganized use of pesticides is of major concern for human health. For this reason, the Stockholm Convention recommends the elimination or restriction of the following compounds: chlordane, DDT, dieldrin, endosulfan, lindane, and pentachlorophenol.

#### 2.4.1. Chlordane

The hypothesis that developmental exposure to chlordane alters intracellular neuronal signalling, contributing to synaptic and behavioral alterations associated with neurodevelopmental disorders, has been tested on organotypic rat hippocampal slices [77]. Low concentrations of chlordane were chronically dosed. Proteomics revealed that chlordane modified the expression of hippocampal proteins involved in pivotal mechanisms in brain development and synaptic signalling, some of which are associated with neurodevelopmental disorders.

#### 2.4.2. DDT

DDT can alter the activities of both antioxidants and glycolytic enzymes. Proteomics was used to investigate the oxidative stress generation in the liver of DDT-fed Mus spretus mice [16]. The data indicated that the liver of exposed mice lacked some protective enzymes, and that DDT caused metabolic reprogramming that increased the glycolytic rate and disturbed lipid metabolism. The results suggested that overall liver metabolism was altered in mice exposed to DDT.

Moreover, Pavlikova et al. found markers of acute toxicity from DDT exposure among proteins expressed in NES2Y human pancreatic beta-cells [7]. The authors found 22 proteins with changed expression, including proteins involved in ER stress, mitochondrial function, and maintenance of cell morphology.

#### 2.4.3. Dieldrin

The effects of dieldrin on the hepatic proteome were determined in zebrafish following dietary treatment [17]. Both label-free and iTRAQ proteomics were conducted. Overall, dieldrin exposures reduced the abundance of proteins associated with glucose and cholesterol metabolism, lipid oxidation, liver function, and immune-related processes. Protein responses in the liver were largely dose-dependent.

Cowie et al. determined the molecular responses in the adult zebrafish central nervous system following dietary exposure to dieldrin [9]. Proteins affected were functionally associated with the mitochondria (ATP synthase subunits, hypoxia up-regulated protein 1, NADH dehydrogenases, and signal recognition particle 9), and a protein network analysis highlighted PD and Huntington’s disease as diseases associated with these altered proteins.

The testis is another organ in which dieldrin can bioaccumulate. A study investigated whether dieldrin, at concentrations within both maternal circulations and environmental ranges, could disrupt the human fetal testis [52]. Dieldrin altered proteins associated with cancer, apoptosis, and development. Dieldrin also reversed some LH-induced changes in protein expression, supporting the conclusion that Leydig cell function is at risk from dieldrin. The results indicate that exposure to dieldrin could affect fetal human Leydig cells and potentially lead to subtle dysregulation of reproductive development and adult fecundity.

#### 2.4.4. Endosulfan

The proteomics signature of endosulfan exposure was reported in HepG2 cells [5]. The results revealed that endosulfan induced significant alterations in the expression level of proteins involved in multiple cellular pathways, such as apoptosis, transcription, immune/inflammatory response, and carbohydrate metabolism.

The effect of a sub-lethal concentration of endosulfan was also investigated on human neuroblastoma cells (SH-SY5Y), using genomic and proteomic approaches [6]. A gene ontology enrichment analysis revealed that the differentially expressed proteins were involved in a variety of cellular events, such as the neuronal developmental pathway, immune response, cell differentiation, apoptosis, transmission of nerve impulse, axonogenesis, etc. Based on the gene and protein profile, possible mechanisms underlying endosulfan neurotoxicity were predicted.

Endosulfan exposure during brain development alters motor activity, coordination, learning, and memory. However, the molecular mechanisms driving these effects have not been studied in detail. In one study, the authors performed a multi-OMICS study in cerebellum on rats exposed to endosulfan during embryonic development [53]. Pregnant rats were orally exposed daily to a low dose of endosulfan. Pathways significantly altered after endosulfan exposure included the phosphatidylinositol signalling system, calcium signalling, the inflammatory and immune system, and protein processing in the endoplasmic reticulum. Sex-dependent effects of endosulfan in the OMICS results that matched sex differences in cognitive and motor tests were found. Indeed, Kern et al. reported a greater susceptibility of males to certain neurotoxicants such as endosulfan [103].

#### 2.4.5. Lindane

Previous studies have reported that lindane exposure induces oxidative stress in the rat brain that may lead to neurodegeneration. We found two studies that aimed to identify the proteins that may be involved in lindane-induced neurotoxicity [78,79]. Data showed that repeated exposure to lindane for 21 days in adult rats significantly increased the reactive oxygen species and lipid peroxidation in different brain regions [78]. An increase in the expression of synuclein in the substantia nigra and corpus-striatum and of the amyloid precursor protein in the hippocampus and frontal-cortex suggested the accumulation of proteins in these brain regions. Western blotting also revealed alterations in the dopaminergic and cholinergic pathways in the hippocampus and substantia nigra isolated from lindane-treated rats. These data suggest that repeated exposure to lindane in adult rats induces similar molecular processes to those observed in neurodegenerative diseases. Similarly, proteomic analysis was carried out in the substantia nigra and hippocampus isolated from rat offspring born to mothers exposed to lindane and subsequently rechallenged at adulthood (12 weeks) [79]. Disruption of proteins related to energy metabolism and oxidative stress were reported, suggesting that prenatal exposure to lindane induces persistent molecular changes in the nervous system of offspring, leading to neurodegeneration following re-exposure in adulthood.

#### 2.4.6. Pentachlorophenol

To further understand the mechanisms of action of pentachlorophenol exposure, proteomic analysis was used to identify proteins differentially expressed in the liver of rare minnow following exposure [21]. Results principally identified proteins involved in transport, metabolism, and oxidative stress response.

To conclude, we noticed that pesticides exert a global neurotoxicity, mediated by a dysregulation of proteins involved in the pathophysiology of neurological disorders (chlordane) and neurodegenerative diseases (dieldrin and lindane). In addition, endosulfan exposure was shown to lead to greater susceptibility of males to neurotoxicity. Studies on potential hepatotoxic effects showed deleterious effects on the overall liver metabolism after DDT and dieldrin exposure, or on calcium signalling after endosulfan and pentachlorophenol exposure. Finally, an impact on the endocrine system was observed, by modifying pancreatic beta-cells proteome (DDT) and in the Leydig cells proteome (dieldrin).

### 2.5. Per- and Polyfluoroalkyl Substances

Per- and polyfluorinated alkyl substances (PFASs) are synthetic chemicals that contain at least one perfluoroalkyl group. Perfluorooctanoic acid (PFOA) and perfluorooctane sulfonate (PFOS) are the most common PFAS detected in the environment due to their widespread use in manufacturing and chemical stability. In recent years, PFAS have been well-studied, as reported in the recent review of Beale et al. [104]. Indeed, many studies illustrated the toxicity of PFAS, mainly reprotoxicity [105], immunotoxicity [106], neurotoxicity [91], and cardiotoxicity [107,108].

#### 2.5.1. PFOA

A proteomic approach was chosen to screen molecular targets affected by PFOA in human HepG2 liver cells [22]. A network analysis revealed proteins that were principally involved in lipid metabolism and cancer. The hepatocyte nuclear factor 4 (HNF4) was the key regulator of the network. Thus, this study provided the first experimental evidence that HNF4 was negatively affected by PFOA. iTRAQ-based quantitative proteomic analysis of rat livers confirmed the impact of PFOA on lipid metabolism mediated by HNF4 [26]. In terms of cellular damage, PFOA exposure was also studied. A study investigated the alterations in protein profile within L-02 cells exposed to PFOA [23]. Induction of apoptosis via the p53-dependent mitochondrial pathway was further suggested as one of the key toxicological mechanisms occurring under PFOA exposure. Conversely, in another study, BALB/C mice were administered PFOA at different doses for 28 days [24]. Liver samples were examined for proteomic changes using iTRAQ labelling. The results showed dose-dependent hepatocyte apoptosis by adenosine receptors via the cAMP pathway. Autophagy was also observed without cell death. In another study, Yan et al. observed autophagosome accumulation in HepG2 cells after PFOA exposure [25]. These findings demonstrated that PFOA blocked autophagy and disturbed intracellular vesicle fusion in the liver. Changes in autophagy were observed only at highly cytotoxic concentrations of PFOA, suggesting that autophagy may not be a primary target or mode of toxicity.

Interestingly, another study found a disturbance of the cAMP pathway mediated by PFOA in SKBR3 cells [13]. Pathway analyses showed that the cAMP signalling pathway was presented as foremost among all the regulatory patterns and concentrations groups of PFOA. After treatment with different concentrations of PFOA, ADORA1 and ADORA2A were respectively activated, showing opposite cellular effects, leading to types of breast lesions. Future therapeutics targeting this pathway could lead to the exertion of potential cell protection in different organs. To study PFOA-induced reprotoxicity, Huang et al. performed a combined proteomics and metabolomics analysis to investigate the alterations in MLTC-1 Leydig cells responsive to low levels of PFOA exposure [65]. The results showed that PFOA regulated the expressions of 18 proteins, and seven metabolites were specifically tied to lipid and fatty acid metabolism, as well as testicular steroidogenesis. It was further suggested that low-dose PFOA stimulated steroid hormone synthesis by accelerating fatty acid metabolism and the steroidogenic process. Chronical doses conferred exposure to different effects. In a study conducted by Zhang et al., male mice were exposed to different doses of PFOA by oral gavage for 28 days [66]. Among the different expressed proteins, INSL3 and CYP11A1, as Leydig-cell-specific markers, were significantly decreased. The findings indicate that PFOA exposure can impair male reproductive function. Further research should investigate both acute and chronic exposure to compare the effects on steroidogenesis.

#### 2.5.2. PFOS

PFOS are known to cause dyslipidaemia in animals and humans. Similarly, PFOS are known to induce hepatic steatosis in animals on a low-fat chow. Different studies have used proteomic analysis to better characterise these mechanisms. By applying iTRAQ labelling quantitative proteomic technology, the liver proteome was analysed in mice exposed to PFOS [27]. Gene ontology analysis showed differentially expressed proteins, mainly involved in lipid metabolism, transport, biosynthetic processes, and stimulus response. The activation of PPARα was a major mechanism in PFOS-mediated cellular responses. Other studies on C57BL/6J mice confirmed the role of PPARα in PFOS-exposure in inducing hepatic steatosis, using proteomic analysis [28], lipidomic and proteomic analysis [32], as well as a transcriptomic and proteomic integrative analysis [33]. An in vivo study exploited multilayered glycoproteomics to quantify the global protein expression levels, glycosylation sites, and glycoproteins in cells exposed to PFOS. They also suggested a role of PPARα in hepatotoxicity caused by PFOS [29]. Li et al. also demonstrated that 16 overexpressed glycoproteins were exclusively related to neutrophil degranulation, cellular stress responses and protein processing in the endoplasmic reticulum. Quantitative proteomic technologies were also applied to in vitro studies in order to investigate the effects of PFOS exposure on hepatic cells. iTRAQ labelling was applied to investigate the differential protein expression profiles of L-02 exposed to PFOS [30]. It was proposed that PFOS contribute to activation of the p53 and c-Myc signalling pathways, which then trigger the apoptotic process. In another experiment, PFOS exposure was analysed in HL-7702 cells. Levels of different cyclin and their partner cdks were elevated [31]. The authors hypothesized that low-dose PFOS stimulates cell proliferation by driving cells into G1 through elevating cyclin/cdk expression.

Developmental effects were also commonly reported. Zhang et al. used embryonic stem cells to assess the developmental cardiotoxicity of PFOS [11]. Pathway analysis revealed that 32 signalling pathways were affected, particularly those involved in metabolism. Changes in five protein levels, including TDH, Xrcc5, SOD2, Dnmt3b, and Dnmt3a were confirmed by Western blotting. These results revealed new potential targets of PFOS in the developmental cardiovascular system. Concerning neurodevelopmental effects, an integrative study (transcriptomics, proteomics and metabolomics) was carried out with zebrafish larvae exposed to different doses of PFOS [12]. Integrated OMICS implied that decisive pathways exist for axonal deformation, neuroinflammatory stimulation, and dysregulation of calcium ion signalling, which are more clearly specified for neurotoxicity.

Overall, studies on PFASs exposure mainly focused on hepatotoxic effects. Proteomic analyses confirmed the role of PPARα in PFOS-mediated cellular responses. The analyses also revealed the role of HNF4 as a key regulator of the network of PFOA-induced hepatotoxic effects. Similarly, both PFAS exposure showed an impact on apoptosis via p53 signalling pathway. Interestingly, PFOA exposure influenced the steroidogenesis, stimulating it in an acute study and impairing it in a chronic study. Finally, only PFOS exposure illustrated developmental effects by proteomic analyses, to assess developmental cardiotoxicity and neurotoxicity.

### 2.6. Synthesis and Limitations

Risk of bias for each included study were assessed by the inclusion and exclusion factors. No missing results were observed on data collected for each study. In contrast, limitations were observed between each study during inclusion and analysis:
-The inability to obtain Mesh terms for each POP.-Physical operator conducted all stages of the screening of articles.-With inclusion for an unlimited period, different kinds of proteomic analyses were reported, depending on the technologies available during the publication of articles.

## 3. Methods

### 3.1. Literature Search

The preferred reporting items for systematic reviews and meta-analyses (PRISMA) method [109] were used for the collection, identification, screening, selection, and analysis of the reviewed studies. This type of approach minimises bias, thus providing reliable results from which to draw conclusions. A literature search was performed using one database: PubMed. The search criteria included scientific articles on the contribution of proteomics to the toxicology of POPs in humans, published by 30 May 2022. The keywords used in the literature search are reported in Appendix A. The selection of POPs was defined according to the definition of the Stockholm Convention [1]. The research question did not involve any specific comparisons. The assessed outcomes were proteomic results. The total number of articles found was 742, which, after sorting out duplicates and reviews, was reduced to 349.

### 3.2. Selection Criteria

In the second screening stage, a total of 178 articles from the initial 349 articles were excluded, based on in-depth observations of the abstracts of the articles. This screening led to the exclusion of records not involving proteomics and/or POPs. Of the 171 articles, the second screening stage permitted further in-depth examination of the full text of all articles. A total of 82 articles were eliminated after the stage 2 screening. Thus, a total of 89 publications were included in the final data collection and analysed further. The systematic article screening stages are shown in Figure 3. To assess the eligibility of the articles, a full-text review was performed by one reviewer (S.G.) and a second reviewer (N.D.) was consulted when necessary. Risk of bias was assessed by the identification of excluded factors.

### 3.3. Data Extraction

During the final selection, data were drawn from 89 studies on POPs. The following features were extracted: molecule, species, contribution of proteomics to POPs toxicology, and technique. No missing results were reported.

## 4. Conclusions

The number of proteomics studies performed in the field of POP research has increased substantially over the past decade. As described above, proteomic approaches have been used to generate results that provide a holistic view of an organism responses to exposure to these chemical substances (Figure 4). Notably, proteomic technologies have generated new data that have led to a better understanding of how these molecules exert their toxicity on their targets. However, the use of proteomic technologies in the field of POP research is still in its embryonic stages and several questions regarding these approaches have not yet been addressed.

First, several chemical substances included in the Stockholm Convention have not been studied by proteomic approaches, as no research articles were found in PubMed for several POPs, such as chlordecone, heptabromodiphenyl ether, hexabromobiphenyl, hexabromodiphenyl ether, hexachlorobenzene, polychlorinated naphtalenes, and toxaphene. Notably, the absence of any proteomic study focusing on chlordecone was quite surprising as this substance is now considered to be a major public health concern in several regions, such as the French West Indies [110].

Secondly, most of the proteomic studies described cellular and molecular responses following exposure to a single molecule. However, humans are continuously exposed throughout their lives to a high number of chemical substances [111,112,113]. Accordingly, future investigations based on proteomic approaches should focus on the long-term accumulation of these substances.

Thirdly, the use of proteomics in the field of toxicology research is quite basic and is currently only focused on the characterisation of protein abundance changes, rarely using high-throughput technologies. Indeed, about 42% of these studies are based on the use of 2D-DIGE and MALDI-TOF MS, whereas only one study describes the use of HDMS. Contrary to LC-MS/MS based approaches, 2D-MALDI-TOF MS approaches do not allow for any in-depth proteome analysis. Therefore, proteomic analysis based on the use of these approaches only study “the tip of the iceberg” of the proteome, leading to a partial comprehension of the cell response to POP exposure. However, because of its robustness, its ability to separate proteoforms, and its easy interface with many powerful biochemistry techniques, 2DE could be an approach of choice if used for the appropriate problem. This concerns the least complex proteomes, such as secretome or interactome, or even analysis of PTMs [114]. In the future, it will be necessary for toxicologists to initiate collaboration with proteomic platforms equipped with latest-generation mass spectrometers and bio-informatic tools, but also use the most appropriate technique to answer the biological question asked.

Moreover, a major concern in proteomic studies is the characterisation of PTMs, such as phosphorylation, glycosylation, and acetylation [115]. However, our literature review identified only two studies that aimed to determine the modification of phosphoproteome [44,45] and glycoproteome [45] after POP exposure. As the PTMs of a protein can determine its activity state, location, turnover, and interactions with other proteins, it is necessary to investigate this using global approaches to better understand how exposure to POPs could modify cell machinery and function. In the same manner, POPs, such as other chemical compounds (naphthalene, bisphenol, etc.) are molecules that could directly interact with proteins in a covalent manner. It was recently showed that furan-containing compounds (FCCs) could covalently interact with proteins in hepatocytes. Indeed, Li et al. identified 171 lysine-based adducted proteins and 145 cysteine-based adducted proteins by the reactive metabolites of the three FCCs [116]. This study highlights that the development of efficient chemoproteomic platform to identify adducted proteins and to predict the toxicity of POPs are a very promising approach, complementary to traditional proteomics.

Finally, proteomics is a single-OMIC analysis that can provide a global view of the response of an organism exposed to a chemical substance, but only by focusing on proteins, and not on other macromolecules (mRNA, carbohydrate, lipid, etc.). It cannot provide a systemic understanding of toxicity pathways or adverse outcome pathways [117]. The next challenge will certainly be the use of multi-OMICS analyses, and their integration in datasets to bring about an improvement in the confidence in detecting the global response/adaptation of an organism to the acute or chronic exposure to distinct POPs. Indeed, each OMICS datum contains specific information that is not present in other data, and multi-OMICS integration can help to provide a more comprehensive overview of the biological response to biotoxin exposure. However, as the different types of OMICS have a large number of heterogeneous biological variables and a low number of biological samples, the incorporation of different biological layers of information to predict phenotypic outcomes remains challenging. To this end, a considerable number of computational tools have been developed over the years [118]. This type of integrative approach will require the establishment of multi-disciplinary collaborations to better characterise the molecular and cellular mechanisms by which POPs exert their toxic effects in humans.

## Figures and Tables

**Figure 1 ijms-23-14271-f001:**
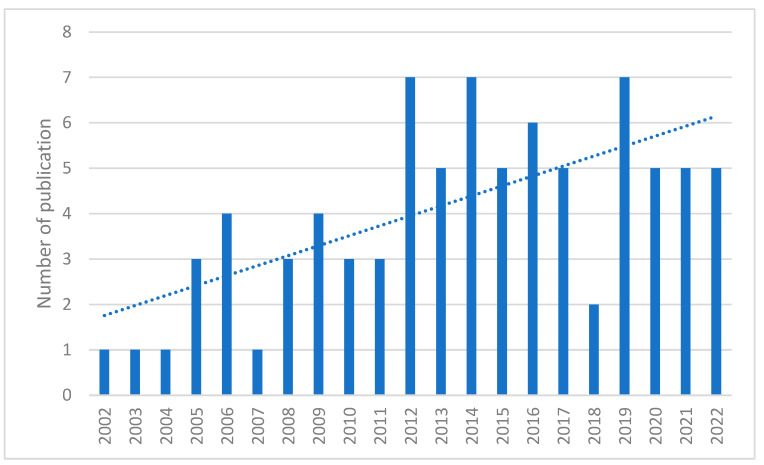
Number of publications on the contribution of proteomics to human toxicology of POPs published each year.

**Figure 2 ijms-23-14271-f002:**
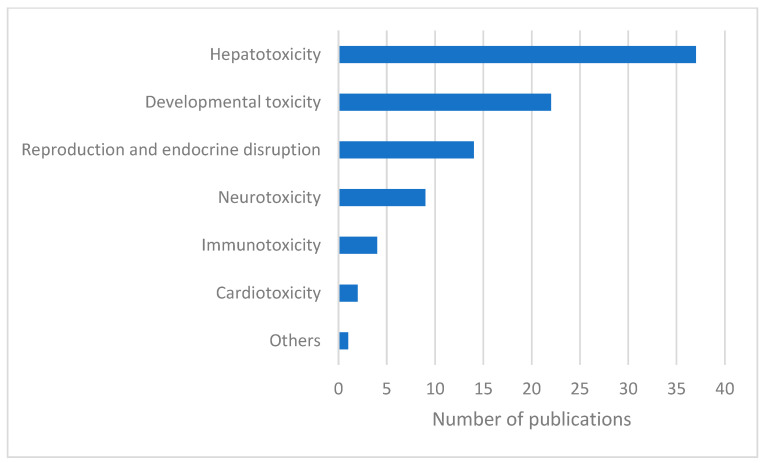
Number of publications on the contribution of proteomics to human toxicology of POPs, classified by toxicity type.

**Figure 3 ijms-23-14271-f003:**
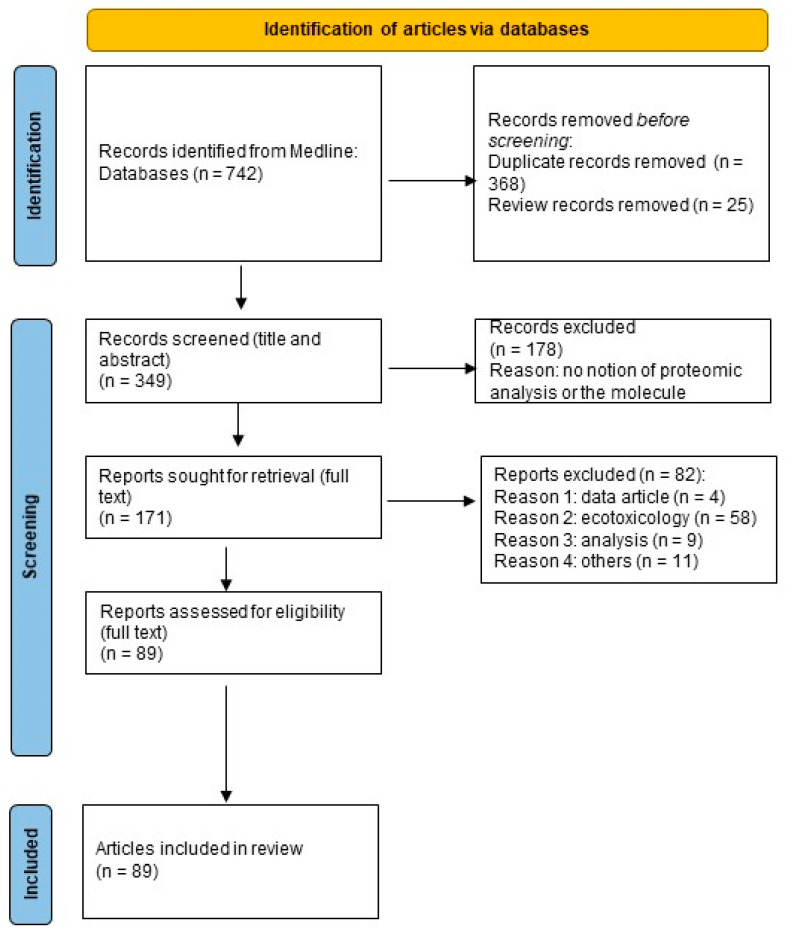
Flowchart of article selection.

**Figure 4 ijms-23-14271-f004:**
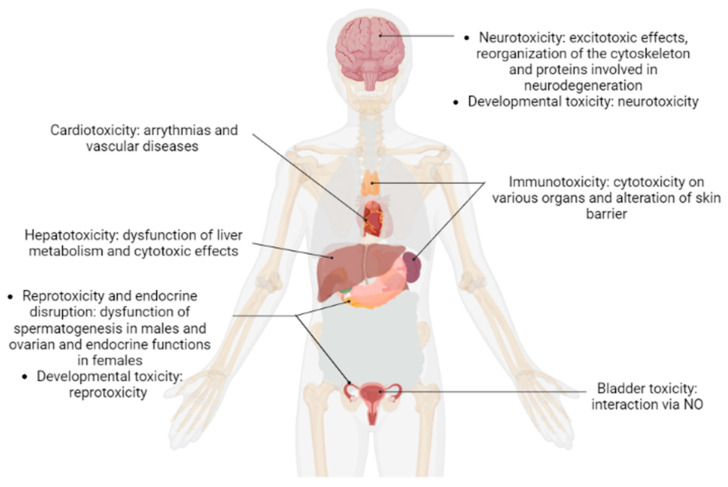
Summary of human toxicity induced by POPs based on proteomic studies.

**Table 1 ijms-23-14271-t001:** Articles describing POP-induced hepatotoxicity.

Molecule	Species	Consequences	Technique	Article
DDT	*Mus spretus* (SPRET/EiJ)	Metabolic reprogramming: impact on energy metabolism (glucose and lipid).	2D-DIGE and MALDI-TOF/TOF	[16]
Dieldrin	*Danio rerio*	Dysregulation of proteins involved in energy metabolism (lipid and glucose), oxidative stress, and drug metabolism.	Label free or iTRAQ labelling and LC-MS/MS	[17]
Endosulfan	*Homo sapiens* cells (HepG2)	Alteration of proteins involved in cellular pathways: apoptosis, immune/inflammatory response, and carbohydrate metabolism.	2D-DIGE and MALDI-TOF	[5]
HBCD	*Danio rerio* cells (liver)	Dysregulated proteins involved in protein metabolism, cellular energy, and cytoskeleton dynamics.	2D-DIGE and MALDI-TOF or FT-ICR	[18]
	*Rattus norvegicus* (Wistar)	Dysregulated proteins involved in oxidative stress response and energy metabolism (glucose, amino acid, and lipid) in females.	2D-DIGE and MALDI-TOF/TOF	[19]
	*Rattus norvegicus* (Wistar)	Dysregulated proteins involved in oxidative stress response and energy metabolism (glucose, amino acid, and lipid), depending on sex.	2D-DIGE and MALDI-TOF/TOF	[20]
Pentachlorophenol	*Gobiocypris rarus*	Dysregulated proteins involved in transport, metabolism, and oxidative stress response.	2D-DIGE and MALDI-TOF/TOF	[21]
PFOA	*Homo sapiens* cells (HepG2)	Dysregulated proteins involved in lipid metabolism and cancer, with HNF4 as the key network regulator.	2D-DIGE and MALDI-TOF	[22]
	*Homo sapiens* cells (L-02)	Impact of apoptosis by activation of p53 pathway.	2D DIGE and MALDI-TOF/TOF	[23]
	*Mus musculus* (BALB/c)	Impact of apoptosis by adenosine receptors (cAMP pathway).	iTRAQ labelling and LC-MS/MS	[24]
	*Homo sapiens* cells (HepG2)	Autophagy blockage and dysregulation of vesicular trafficking.	iTRAQ labelling and LC-MS/MS	[25]
	*Rattus norvegicus* (Sprague-Dawley)	Impact of lipid metabolism mediated by HNF4.	iTRAQ labelling and LC-MS/MS	[26]
PFOS	*Mus musculus* (Kunming)	Involvement of PPARα mechanism in cellular effects.	iTRAQ labelling and nanoLC-MS/MS	[27]
	*Mus musculus* (C57BL/6J)	Activation of PPARα: proteins involved in fatty acid metabolism, lipid synthesis, and xenobiotic metabolism.	SWATH-MS	[28]
	*Mus musculus* (C57BL/6J)	Glycoproteomics: activation of PPARα mechanism and dysregulation of proteins involved in ER stress and neutrophil degranulation.	nanoLC-MS/MS	[29]
	*Homo sapiens* cells (L-02)	Triggering of the apoptotic process by activating of the p53 and c-Myc signalling pathways.	iTRAQ labelling and 2D nanoLC-MS/MS	[30]
	*Homo sapiens* cells (HL-7702)	Upregulation of proteins driving cells into the cell cycle from the G1 stage.	iTRAQ labelling and LC-MS/MS	[31]
	*Mus musculus* (C57BL/6J)	Lipidomic analysis: activation of PPARα mechanism.	SWATH-MS	[32]
	*Mus musculus* (C57BL/6J)	Transcriptomic and proteomic integrative analysis: activation of PPARα mechanism.	SWATH-MS	[33]
PCB	*Mus musculus* (C3H/N)	Dose-dependent increases in Cyp1a1 and Cyp2b10.	2D-DIGE and MALDI-TOF/TOF	[34]
	*Mus musculus* (C57BL/6J)	Dysregulation of transcription factors involved in liver function.	LC-MS/MS	[35]
	*Mus musculus* (C57BL/6J)	Overall impact on liver metabolism by activating AhR.	TMT labelling and nanoLC-MS/MS	[36]
	*Mus musculus* (C57BL/6J)	Ahr-dependent mechanism: dysregulation of proteins involved in liver metabolism and bioenergetics and endocrine response and function.	TMT labelling and nanoLC-MS/MS	[37]
PCDD	*Rattus norvegicus* (Sprague-Dawley)	Short-term exposure: increase of apolipoprotein A-IV.	2D-DIGE and nanoLC-MS/MS	[38]
	*Rattus norvegicus* cells (H4IIEC3)	Dysregulated proteins involved in oxidative stress and mitochondrial function.	2D-DIGE and MALDI-TOF/TOF	[39]
	*Rattus norvegicus* (Sprague-Dawley)	Dysregulation of proteins involved in the NF-κB pathway, which may increase oxidative stress.	2D DIGE and MALDI-TOF	[40]
	*Rattus norvegicus* cells (H4IIEC3)	Dysregulated proteins involved in cell cycle regulation, growth factor signalling, and control of apoptosis.	ICPL labelling and 1-D nano-LC and MALDI-TOF/TOF	[41]
	*Callithrix jacchus*	Dysregulation of proteins involved in the immune response and downregulation of thymidine phosphorylase.	2D DIGE and LC-MS/MS	[42]
	*Rattus norvegicus* (Sprague-Dawley)	Fibrinopeptide A levels correlated with liver fatty degeneration and necrosis.	magnetic bead separation and MALDI-TOF/TOF and nanoLC-MS/MS	[43]
	*Rattus norvegicus* cells (H4IIEC3)	Phosphoproteomics: dysregulation of transcription factors (ARNT) and coregulators and regulators of small GTPases.	SILAC labelling and LC-MS/MS	[44]
	*Homo sapiens* cells (Chang liver)	Dysregulated proteins involved in cellular protein folding and turnover, energy metabolism, cytoskeletal network, and vesicular trafficking.	2D-DIGE and MALDI-TOF/TOF	[45]
	*Rattus norvegicus* (Long evans and Han/Wistar)	Role of AhR in major differences in response between strains of rats.	2D DIGE and LC-MS/MS	[46]
	*Mus musculus* (C57BL/6 and DBA/2 J)	Three altered proteins (SNRK, IGTP, and IMPA2) showing consistent strain-dependent changes.	LC-MS (HDMS)	[15]
	*H* *omo sapiens*	Seven upregulated proteins: AFP, fibronectin, pre-albumin, fibrinogen gamma A chain precursor, XAP-5, human rab GDI, and follistatin. One downregulated protein: albumin.	2D-DIGE and MALDI-TOF/TOF	[47]
	*Homo sapiens*	Upregulation of proteins involved in oxidative stress.	2D-DIGE and MALDI-TOF	[48]
	*Homo sapiens* cells (HepG2)	Validation of eight biomarkers (GLO 1, HGD, PRX 1, PSMB 5 and 6, UDPGlcDH, HADH, and STF).	2D-DIGE and nanoLC-MS/MS	[49]
PCDF	*Homo sapiens* cells (HepG2)	Potential biomarkers: protein DJ-1, proteasome activator complex subunit 1, and plasminogen activator inhibitor-3.	2D-DIGE and nanoLC-MS/MS	[50]

**Table 2 ijms-23-14271-t002:** Articles describing POP-induced developmental toxicity.

Molecule	Species	Consequences	Technique	Article
BDE-209	*Rattus norvegicus* cells (neural stem culture)	Decrease in proteins involved in cytoskeletal maintenance.	2D-DIGE and MALDI-TOF	[51]
Dieldrin	*Homo sapiens* culture (fetal testis)	Dysregulation of proteins involved in the incidence of abnormal cell development: antagonization of LH effects.	2D-DIGE and MALDI-TOF	[52]
Endosulfan	*Rattus norvegicus* (Wistar)	In males: alteration of motor coordination through phosphatidylinositol pathways. In females: up-regulation of GAD.	TMT LC-MS3	[53]
HBCD	*Mus musculus* (BALB/c)	Dysregulated proteins involved in the disruption of cellular calcium homeostasis.	2D-DIGE and LC-MS/MS	[54]
	*Mus musculus* (BALB/c)	Alteration of T/E2 ratios through calcium dysfunction, responsible for oxidative stress and lipid metabolism.	2D-DIGE and LC-MS/MS	[55]
BDE-99	*Mus musculus* (NMRI)	Altered proteins in the striatum involved in neurodegeneration and neuroplasticity; in the hippocampus, proteins involved in metabolism and energy production were affected.	2D-DIGE and MALDI-TOF-MS	[56]
PCB	*Ovis aries* (Norwegian White)	Subtle effects on the developing fetal testis: changes in cellular processes as stress response and cytoskeleton regulation.	2D-DIGE and LC-MS/MS	[57]
	*Rattus norvegicus* neurons (Wistar)	Dysregulation of proteins involved in calcium homeostasis, cytoskeletal process, and trafficking of proteins.	LC-MS/MS	[58]
	*Rattus norvegicus* (Wistar)	Dysregulation of proteins involved in calcium homeostasis and phosphoinositol signalling pathway.	1D-DIGE and LC-MS/MS	[59]
	*Rattus norvegicus* (Long-Evans)	Dysregulation of proteins involved in calcium homeostasis.	2D-DIGE and MALDI-TOF/TOF	[60]
	*Mus musculus* (BALB/c)	Dysregulation of proteins involved in calcium homeostasis and lipid metabolism.	2D-DIGE and LC-MS/MS	[54]
	*Mus musculus* (BALB/c)	Dysregulation of proteins involved in calcium homeostasis and energy metabolism.	2D-DIGE and LC-MS/MS	[55]
PCDD	*Gallus gallus domesticus* (chick)	Dysregulated proteins involved in blood clotting, oxidative stress, electron transport, and calcium regulation.	2D-DIGE and LC-MS	[61]
	*Mus musculus* (BALB/c)	Probable proliferation of corneal epithelial cells before eye opening: increase in stathmin 1.	2D-DIGE and MALDI-TOF/TOF	[62]
	*Mus musculus* (C57BL/6J)	Association peroxiredoxin-1 and cleft palate formation.	2D-DIGE and MALDI-TOF/TOF	[63]
	*Mus musculus* (BALB/c)	Dysregulation of proteins involved in calcium homeostasis and lipid metabolism.	2D-DIGE and LC-MS/MS	[54]
	*Mus musculus* (BALB/c)	Dysregulation of proteins involved in calcium homeostasis and energy metabolism.	2D-DIGE and LC-MS/MS	[55]
	*Danio rerio*	Role in neurological pathway: WFIKKN1 involved in skeletal muscle development.	LC-MS	[64]
BDE-47	*Rattus norvegicus* cells (neural stem culture)	Decrease in proteins involved in cytoskeletal maintenance.	2D DIGE and MALDI-TOF-MS	[51]
	*Mus musculus* (BALB/c)	Dysregulation of proteins involved in calcium homeostasis and lipid metabolism.	2D-DIGE and LC-MS/MS	[54]
PFOS	*Mus musculus* cells (embryonic stem)	Cardiovascular dysfunction: dysregulation of proteins involved in xenobiotic metabolism.	SILAC labelling and LC-MS/MS	[11]
	*Danio rerio*	Abnormal neurological development: dysregulation of Rab proteins and proteins involved in calcium homeostasis and axonal formation.	TMT labelling and LC-HR-MS	[12]

**Table 3 ijms-23-14271-t003:** Articles describing POP-induced reprotoxicity and endocrine disruption.

Molecule	Species	Consequences	Technique	Article
PFOA	*Mus musculus* cells (MLTC-1 Leydig)	Stimulation of steroidogenesis by accelerating fatty acid metabolism and steroidogenic processes.	nanoLC-MS/MS	[65]
	*Homo sapiens* cells (SKBr3)	Dysregulation of the cAMP pathway by adenosine receptors.	TMT labelling and nanoLC-MS/MS	[13]
	*Mus musculus* (BALB/c)	Inhibition of proteins involved in steroidogenesis (CYP11A1, INSL3) in a dose-dependent manner.	iTRAQ labelling and LC-ESI-MS/MS	[66]
PCB	*Homo sapiens* cells (MCF7)	Endocrine disruption properties: dysregulated proteins involved in stress response and cytoskeleton process.	2D-DIGE and MALDI-TOF/TOF	[67]
	*Homo sapiens* cells (MCF7)	Endocrine disruption properties: dysregulated proteins involved in stress response and cytoskeleton process in subcellular fractions.	2D-DIGE and MALDI-TOF/TOF	[68]
	*Homo sapiens* cells (H295R)	Perturbation of steroidogenesis: dysregulated proteins involved in protein synthesis, cortisol synthesis, stress response, and apoptosis.	2D-DIGE and LC-MS/MS	[69]
	*Mus musculus* gland organoids (FVB/n)	Oestrogen-independent effects: dysregulation of proteins involved in apoptosis, cell adhesion, and proliferation.	iTRAQ labelling and LC-MS/MS	[70]
PCDD	*Sus domesticus* cells (granulosa from AVG-16 line)	Affects ovarian follicle fate by reorganising the cytoskeleton and extracellular matrix and modulating proteins important for the cellular response to stress.	2D-DIGE and MALDI-TOF/TOF	[71]
	*Sus domesticus* cells (granulosa from AVG-16 line)	Alteration of three proteins by an AhR-independent mechanism: PDI, ATPbeta, and AnxA5.	2D-DIGE and MALDI-TOF/TOF	[72]
	*Rattus norvegicus* (Sprague-Dawley)	Upregulation of six testicular proteins involved in stress response and downregulation of SP22, a potential biomarker to diagnose human infertility in dioxin-exposed men.	2D-DIGE and MALDI-TOF/TOF	[73]
	*Rattus norvegicus* (Sprague-Dawley)	Disruption of ovarian and endocrine functions by dysregulated proteins involved in stress response.	2D-DIGE and MALDI-TOF/TOF	[74]
BDE-47	*Rattus norvegicus* (Sprague-Dawley)	Disruption of spermatogenesis: dysregulated proteins involved in apoptosis and mitochondrial function.	2D-DIGE and MALDI-TOF/TOF	[75]
	*Mus musculus* (C57BL/6J)	Affects male reproduction: dysregulation of proteins involved in stress response, cell maintenance, and inflammatory response.	iTRAQ labelling and nanoLC-MS/MS	[76]
DDT	*Homo sapiens* cells (NES2Y)	Dysregulated proteins involved in stress response, mitochondrial process, cytoskeletal process, and cell maintenance.	2D-DIGE and MALDI-TOF/TOF	[7]

**Table 4 ijms-23-14271-t004:** Articles describing POP-induced neurotoxicity.

Molecule	Species	Consequences	Technique	Article
Chlordane	*Rattus norvegicus* (Wistar)	Dysregulation in the hippocampus of proteins involved in brain development and synaptic signalling.	LC–MS/MS	[77]
Dieldrin	*Danio rerio*	Dysregulated proteins involved in Parkinson’s and Huntington’s disease and mitochondria function.	iTRAQ and LC-MS/MS	[9]
Endosulfan	*Homo sapiens* cells (SH-SY5Y)	Dysregulated proteins involved in regulation of neuronal development, cell adhesion and related processes, apoptosis, signal transduction, and regulation of transmission of nerve impulses.	2D-DIGE and MALDI-TOF	[6]
Lindane	*Rattus norvegicus* (Wistar)	Accumulation of lewy body and beta-amyloid fibrils in the brain, dysfunction of neurotransmission, and dysregulated proteins involved in energy metabolism and oxidative stress.	2D DIGE and MALDI-TOF/TOF	[78]
	*Rattus norvegicus* (Wistar)	Dysregulation of proteins related to energy metabolism and oxidative stress.	2D DIGE and MALDI-TOF/TOF	[79]
BDE-99	*Rattus norvegicus* culture cells (Sprague-Dawley)	Disruption of neurite growth: dysregulation of cytoskeletal proteins and Gap43.	2D-DIGE and nanoLC-MS/MS	[80]
BDE-47	*Mus musculus* (C57BL/6J)	Dysregulation of proteins involved in the Parkinson’s disease pathology.	iTRAQ and nano LC-MS	[81]
	*Rattus norvegicus* (Sprague-Dawley)	Affection of protein degradation pathways: neurotransmitter system disturbance, increased formation of a-synuclein aggregate, and induction of motor defect, all hallmarks of Parkinson’s disease.	LC-MS/MS	[82]

**Table 5 ijms-23-14271-t005:** Articles describing POP-induced immunotoxicity.

Molecule	Species	Consequences	Technique	Article
PCDD	*Rattus norvegicus* (Sprague-Dawley)	Stimulation of immune system: increase in cytokeratin 8 polypeptide, Ig lambda-1 chain C region, and Ig lambda-2 chain C region.	2D-DIGE and MALDI-TOF or nanoLC-MS/MS	[83]
	*Callithrix jacchus*	Dysregulation of proteins involved in oxidative stress and cytoskeleton maintenance.	2D-DIGE and LC-MS/MS	[42]
	*Rattus norvegicus* cells (MSC)	Affects osteoblast differentiation by altering their cell architecture, adhesive properties, and calcium homeostasis.	2D-DIGE and LC-MS/MS	[84]
	*Homo sapiens* cells (hEpB, hEpE and SIK)	Decreases in the differentiation keratinocyte markers: filaggrin, keratin 1, and keratin 10.	2D-DIGE and LC-MS/MS	[85]

**Table 6 ijms-23-14271-t006:** Articles describing POP-induced cardiotoxicity.

Molecule	Species	Consequences	Technique	Article
PCDD	*Danio rerio*	Cardiac hypertrophy and heart failure: alterations of proteins involved in calcium handling, energy metabolism, and cellular redox state.	2D-DIGE and LC-MS/MS	[86]
PCB	*Homo sapiens*	Positive association of PCB exposure: proteins involved in lipid metabolism. Negative association: proteins involved in inflammation/immunity response and platelet degranulation.	nanoLC-MS/MS	[87]
	*Homo sapiens* cells (HMVEC)	Vascular wall damage by a central network focused on c-Myc and TGFB1.	2D-DIGE and MALDI-TOF/TOF	[88]

**Table 7 ijms-23-14271-t007:** Articles describing other POP-induced toxicities.

Molecule	Species	Consequences	Technique	Article
PCDD	*Homo sapiens* cells (RT4)	PCDD exposure: interaction between calcium and iron via NO upon PCDD exposure.	2D-DIGE and MALDI-TOF	[89]

## Data Availability

Not applicable.

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
