# Peer review of "Studying the Impact of Persistent Organic Pollutants Exposure on Human Health by Proteomic Analysis: A Systematic Review"

_ijms, 2022, doi:10.3390/ijms232214271_

Round 1

Reviewer 1 Report

Line 20: Please change the sentence to, "Of the 742 items originally identified, 89 were considered.".
Lines 29-31: Three sentences in a row begin with "they", please change.
Line 32: Please insert "over time" after "living beings".

Line 36: Instead „of a multitude of species and human beings” write “of a variety of animal species and humans”.

Lines 38 and 39: These two sentences contain the same statement as some in the previous paragraph, please merge them.

Lines 41 and 42: Please change to “could be the cause of the development of endocrine, metabolic, immunological, and neurological pathologies”.

Line 45: Please, change “particularly polluting” with more appropriate wording.

Line 59: Please change “substance in Human” into “substances in humans”.

Line 65: Please change existing wording to ”characterize the human health consequences of POP exposure”.

Line 82: Please next to “pI” in the text insert “(charge)”.

Line 88: I do not understand “without a priori” what, knowledge?

Line 101: Please change “to” to “in”.

Line 108: What does “MS/MS sampling rate” stand for? Identification rate?

Line 124: Please put “of POPs” after the “toxicology” and “by” instead of “up to”.

Line 135: Please put “were eliminated” instead of “remained”, or similar, as this form is misleading.

Line 143: This should be Table 1; my suggestion is that instead of “study” it should say “articles” (also to be changed in the figure), but not mandatory.

Line 144: Is it correct 90 or 89?

Line 149: Please check the part of the sentence “through 2022”, did you mean “from 2002 to 2022” or something else?

Line 149: Change “Figure 1” to “Figure 2”.

Line 150: “Additional” to what? Did you mean something like “more attention”?

Line 153: Change “Figure 1” to “Figure 2”.

Line 156: Change “Figure 2” to “Figure 3”.

Line 159: Change “Figure 2” to “Figure 3”.

Line 161: The study with reference number 20 was done only on female rats, and in Table 1 it says "gender dependence", could you check this statement? Perhaps the reference numbers were confused, since the study with reference number 21 deals with both genders?

Line 191: This sentence refers only to BALB mice, but the reference 82 is the study on SD rats.

Line 327: Please delete “by”.

Line 332: Please change “administrated” to “administered”.

Line 333: Please delete “by”.

Lines 350 to 352: Please change all “wfikkn” to “WFIKKN”.

Line 388: Please write “dysregulated” instead of “deregulated”.

Line 506: Please insert “of” between “activities” and “both”.

Line 508: Please write “Mus” instead of “mus”.

Line 574: Maybe to differ from the original study to use “re-exposure” instead of “rechallenge”?

Line 603: Please delete the full stop after “exposure”.

If authors agree maybe to add (where missing and expand a bit where present) a short paragraph (up to ten sentences) at the end of each major group of toxins summarizing the main facts found in studied literature.

Author Response

Reviewer 1:

We are grateful for carefully reading our manuscript. We thank the reviewer 1 for valuable comments and answered to him below.

  • Line 20: Please change the sentence to, "Of the 742 items originally identified, 89 were considered."

We corrected the sentence.

  • Lines 29-31: Three sentences in a row begin with "they", please change.

We replaced "they" with other words

  • Line 32: Please insert "over time" after "living beings".

We inserted “over time” after “living beings”.

  • Line 36: Instead „of a multitude of species and human beings” write “of a variety of animal species and humans”.

We replaced “of a multitude of species and human beings” with “of a variety of animal species and humans”.

  • Lines 38 and 39: These two sentences contain the same statement as some in the previous paragraph, please merge them.

We replaced the two sentences with “Thus, the human population is exposed daily to chemical pollutants through the environment and food, which accumulate in all living beings, particularly in adipose tissue due to their predominantly lipophilic properties.”

  • Lines 41 and 42: Please change to “could be the cause of the development of endocrine, metabolic, immunological, and neurological pathologies”.

We replaced the sentence with “could be the cause of the development of endocrine, metabolic, immunological, and neurological pathologies”.

  • Line 45: Please, change “particularly polluting” with more appropriate wording.

We replaced the expression with “of particular concern for human health”.

  • Line 59: Please change “substance in Human” into “substances in humans”.

We replaced the expression “substance in Human” with “substances in humans”

  • Line 65: Please change existing wording to ”characterize the human health consequences of POP exposure”.

We replaced with “to better characterize the human health consequences of POPs exposure”.

  • Line 82: Please next to “pI” in the text insert “(charge)”.

We inserted “(charge)” next to pI.

  • Line 88: I do not understand “without a priori” what, knowledge?

A priori refers to a kind of proteomics using a global bottom up approach, contrary to a targeted approach.

  • Line 101: Please change “to” to “in”.

We replaced “to” with “in”.

  • Line 108: What does “MS/MS sampling rate” stand for? Identification rate?

We added an explanation “(the rate of signal sampling within a definite time)”.

  • Line 124: Please put “of POPs” after the “toxicology” and “by” instead of “up to”.

We did the two modifications.

  • Line 135: Please put “were eliminated” instead of “remained”, or similar, as this form is misleading.

We replaced “remained” with “were eliminated”.

  • Line 143: This should be Table 1; my suggestion is that instead of “study” it should say “articles” (also to be changed in the figure), but not mandatory.

We replaced “Figure 3” with “Figure 1” in the manuscript and “study” with “articles”.

  • Line 144: Is it correct 90 or 89?

We replaced 90 with 89.

  • Line 149: Please check the part of the sentence “through 2022”, did you mean “from 2002 to 2022” or something else?

We replaced “through 2022” with “up to 2022”.

  • Line 149: Change “Figure 1” to “Figure 2”.

We replaced “Figure 1” with “Figure 2”.

  • Line 150: “Additional” to what? Did you mean something like “more attention”?

We replaced “additional” with “more attention”.

  • Line 153: Change “Figure 1” to “Figure 2”.

We replaced “Figure 1” with “Figure 2”.

  • Line 156: Change “Figure 2” to “Figure 3”.

We replaced “Figure 2” with “Figure 3”.

  • Line 159: Change “Figure 2” to “Figure 3”.

We replaced “Figure 2” with “Figure 3”.

  • Line 161: The study with reference number 20 was done only on female rats, and in Table 1 it says "gender dependence", could you check this statement? Perhaps the reference numbers were confused, since the study with reference number 21 deals with both genders?

We corrected the mistake. We added “in females” for the reference 20 and we added “with gender dependence” for the reference 21.

  • Line 191: This sentence refers only to BALB mice, but the reference 82 is the study on SD rats.

We corrected the mistake. We modified the sentence to “in mice [81] and rats [82]”.

  • Line 327: Please delete “by”.

We deleted “by”.

  • Line 332: Please change “administrated” to “administered”.

We replaced “administrated” with “administered”.

  • Line 333: Please delete “by”.

We deleted “by”.

  • Lines 350 to 352: Please change all “wfikkn” to “WFIKKN”.

We replaced all “wfikkn” with “WFIKKN”.

  • Line 388: Please write “dysregulated” instead of “deregulated”.

We replaced “dysregulated” with “deregulated”.

  • Line 506: Please insert “of” between “activities” and “both”.

We inserted “of” between “activities” and “both”.

  • Line 508: Please write “Mus” instead of “mus”.

We replaced “mus” with “Mus”.

  • Line 574: Maybe to differ from the original study to use “re-exposure” instead of “rechallenge”?

We replaced “rechallenge” with “re-exposure”.

  • Line 603: Please delete the full stop after “exposure”.

We deleted the full stop after “exposure”.

  • If authors agree maybe to add (where missing and expand a bit where present) a short paragraph (up to ten sentences) at the end of each major group of toxins summarizing the main facts found in studied literature.

We added a short paragraph at the end of each major group of toxins:

BFRs: Lines 254-264: “To conclude this part, proteomic studies on neuronal cells highlighted that BFRs exert similar neurotoxicological outcomes. Notably, these results suggest that BDE-47 could be a potential neurotoxicant involved in the pathophysiology of PD. Regarding neurodevelopmental effects, several dysregulated proteins were found to be similar after BFRs exposures: cofilin-1 and vimentin, cytoskeletal proteins, and proteins involved in calcium homeostasis. In contrast, the cellular mechanisms involved in reprotoxicity were heterogenous: BDE-47 exposure lead to the dysregulation of proteins involved in mitochondrial function and oxidative stress, whereas BDE-99 exposure modify the expression of proteins involved in cytoskeleton maintenance. Hepatotoxic effects were analysed by proteomic techniques only on HBCD. An overall decrease in the abundance of protein involved in metabolic processes was noted.”

Dioxins: Lines 420-431: “Taken together, the results demonstrated a wide range of cellular changes after PCDD exposure on different organs. The role of AhR in PCDD-mediated toxicity was showed in studies regarding hepatotoxic, reprotoxic and developmental effects. Studies that aim to evaluate hepatotoxic effects also showed a dysregulation of proteins involved in metabolic processes (especially the blood coagulation pathway and lipid metabolism) and oxidative stress. Interestingly, SP22 was suggested as a potential biomarker of the reprotoxic effects of PCDD. Surprisingly, exposure to PCDF was poorly described by proteomic techniques, and only one study on hepatotoxic effects was reported.” Finally, a study on cardiotoxic effects indicated that exposure to PCDD could lead to heart disease.

PCBs: Lines 506-512: “Together, proteomic approaches revealed a variety of molecular effects following cell exposure to PCBs. As showed with PCDD, a role for AhR in PCBs-mediated toxicity was showed in studies regarding hepatotoxic effects. Developmental effects studies showed subtle effects for PCBs on the proteome of the fetal testes but not on morphology and testosterone production, whereas endocrine studies demonstrated an impact of PCBs on steroidogenesis. Interestingly, proteomic studies on brain development highlighted a premature ageing after exposure. Finally, PCB exposure may increase the risk of cardiovascular disease by affecting lipid metabolism and vessel formation.”

Pesticides: Lines 603-310: “To conclude, we noticed that pesticides exert a global neurotoxicity, mediated by a dysregulation of proteins involved in the pathophysiology of neurological disorders (chlordane) and neurodegenerative diseases (dieldrin and lindane). Also, endosulfan exposure was shown to lead to a greater susceptibility of males to neurotoxicity. Studies on potential hepatotoxic effects showed deleterious effects on the overall liver metabolism after DDT and dieldrin exposure, or on calcium signaling after endosulfan and pentachlorophenol exposure. Finally, an impact on the endocrine system was observed, by modifying pancreatic beta-cells proteome (DDT) and in the Leydig cells proteome (dieldrin).”

PFAS: Lines 693-700: “Overall, studies on PFASs exposure focused on hepatotoxic effects. Proteomic analyses confirmed the role of PPARα in PFOS-mediated cellular responses. The analyses also revealed the role of HNF4 as a key regulator of the network of PFOA-induced hepatotoxic effects. Similarly, both PFAS exposure showed an impact on apoptosis via p53 signalling pathway. Interestingly, PFOA exposure influenced the steroidogenesis, stimulating it in an acute study and impairing it in a chronic study. Finally, only PFOS exposure illustrated developmental effects by proteomic analyses, to assess developmental cardiotoxicity and neurotoxicity. “

Reviewer 2 Report

In the manuscript ijms-202816, Guillotin and Delcourt review the proteomic literature on the effects of persistent organic pollutants (POPs). While the overall topic is of high interest and the reviewing effort appreciated, in my humble opinion the manuscript is very flat. In other words it lacks perspective and does not give the reader the keys to really appreciate and measure the input of proteomics in the field. Moreover, some polishing is necessary to improve the quality of the manuscript.

I will start this review by the latter point. I perfectly understand the purpose of the tables before the main “ results “ text, i.e. to provide some kind of double entry process either by compound or by type of toxic effect. This should be clearly stated, as it would greatly help the reader. Moreover, the tables need serious reviewing, as they are plagued by expressions that have little sense, if any? I have highlighted a few here below, but I may have missed some:

“ Warburg metabolic “ (effect I suppose)

" Dose-response effect with PPARα mechanism " 

" Clear-cut of biological response (glycolysis, inflammation) "

"A subset of hepatic proteins differentially and specifically regulated by AhR"

"Dysregulated proteins involved in oxidative stress, in the mitochondrial function which of VDAC2”

"Presence of NF-jB pathway members which may increase of oxidative stress" NF-kB I suppose

In the text itself in line 44 it is not " aims to eliminate "  but " aimed at  eliminating"? Same problem with the verb “aim” in line 68

Line 350 “wfikkn1’s role” is a bit hard for the reader “wfikkn1 protease inhibitor role” might be clearer.

Moving on the perspective issue, the current manuscript presents the same weakness that most of the proteomic papers, in the sense that it is very descriptive. The key issue in proteomics is not much the analysis depth, but the biological predictive value of the changes observed in a proteomic screen (and in any omic screen indeed). This seems to be quite an unsaid taboo in omics in general and in proteomics in particular, as I know of only one paper that tackles this issue frontally (doi:10.1016/j.jprot.2021.104178), with quite mixed results indeed. It would therefore be great if the authors could highlight the few original research papers cited in their review (if any) that did perform any kind of validation studies to confirm the proteomic results. I understand that this may be more difficult for people working on animal models, but there is little excuse for the people working on cell lines such as HepG2.If there are no papers with validation experiments, then it must be mentioned.

In the same trend the authors complain about the wide use of 2D DIGE in the literature that they surveyed and of its lack of depth (which is true) but forget to mention the advantages of this type of approach especially in terms of PTMs. The use of 2D-based proteomics compared to shotgun has been reviewed in many papers. I can propose two of them (doi: 10.1016/j.jprot.2014.03.035 and doi:10.3390/proteomes8030017), but the authors may choose among the many others present in the literature.

To end with this type of issues, I strongly disagree with the authors regarding their final conclusion (lines 712ff), and I will explain why. Because of the multilayered regulations present in living cells, omic strategies provide molecular evidences, but only biological hypotheses. Consequently, piling omics (as the authors suggest) just piles hypotheses, but does not provide biological evidences. It may just reduce the uncertainty in the translation from molecular events to biological consequences. This problem is especially true for transcriptomics and shotgun proteomics, which completely ignore PTMs and even translational regulations for the former (for the authors’ information, no citation needed here, a discussion of this specific point can be found here: doi: 10.3390/proteomes8030023).

So instead of the complaint on the lack of depth and of the ungrounded (in my opinion) hope in multi-omics, which have been read dozens of times, I suggest that the authors orient their discussion toward more unsung challenges, such as the validation issues (critical in my eyes) and the specific scientific questions risen if we want to consider and tackle potential direct protein modifications by POP-derived reactive metabolites (if this is relevant, owing to the poor chemical reactivity of POPs), as described for other types of pollutants (e.g. in doi: 10.1016/j.toxlet.2013.03.031, doi: 10.1021/tx6003166, doi: 10.1021/tx3002675, doi: 10.1093/toxsci/kfs005, doi: 10.1002/pmic.200401278).

Author Response

Reviewer 2:

We are grateful for carefully reading our manuscript. We thank the reviewer 2 for valuable comments and answered to him below.

In the manuscript ijms-202816, Guillotin and Delcourt review the proteomic literature on the effects of persistent organic pollutants (POPs). While the overall topic is of high interest and the reviewing effort appreciated, in my humble opinion the manuscript is very flat. In other words it lacks perspective and does not give the reader the keys to really appreciate and measure the input of proteomics in the field. Moreover, some polishing is necessary to improve the quality of the manuscript.

  • I will start this review by the latter point. I perfectly understand the purpose of the tables before the main “ results “ text, i.e. to provide some kind of double entry process either by compound or by type of toxic effect. This should be clearly stated, as it would greatly help the reader.

To be clear on this point, we added the following paragraph “Articles addressing the contribution of proteomics to the human toxicology of POPs were categorized according to the different types of toxicity they describe (Figure 3, Table 1-7). We then proceeded in the reverse direction, describing each molecule in the manuscript according to the type of toxicity.”

  • Moreover, the tables need serious reviewing, as they are plagued by expressions that have little sense, if any? I have highlighted a few here below, but I may have missed some:

To be more readable, we improved expressions in the different tables.

  • “ Warburg metabolic “ (effect I suppose)

We replaced the sentence with “Metabolic reprogramming: impact on energy metabolism (glucose, lipid).”

  • " Dose-response effect with PPARα mechanism " 

We replaced the sentence with “Involvement of PPARα mechanism in cellular effects.”

  • " Clear-cut of biological response (glycolysis, inflammation) "

We replaced "Clear-cut of biological response (glycolysis, inflammation) " with “Dose-dependent increases in Cyp1a1 and Cyp2b10.”

  • "A subset of hepatic proteins differentially and specifically regulated by AhR"

We replaced "A subset of hepatic proteins differentially and specifically regulated by AhR" with “Role of AhR in major differences in response between strains of rats.”

  • "Dysregulated proteins involved in oxidative stress, in the mitochondrial function which of VDAC2”

We replaced "Dysregulated proteins involved in oxidative stress, in the mitochondrial function which of VDAC2” with “Dysregulated proteins involved in oxidative stress and in the mitochondrial function.”

  • "Presence of NF-jB pathway members which may increase of oxidative stress" NF-kB I suppose

 We replaced "Presence of NF-jB pathway members which may increase of oxidative stress" with “Dysregulation of proteins involved in the NF-κB pathway which may increase oxidative stress.”

  • In the text itself in line 44 it is not " aims to eliminate "  but " aimed at  eliminating"? Same problem with the verb “aim” in line 68

We replaced “aims to eliminate” with “aimed at eliminating” line 44, and “aims to profile” with “aims at profiling” line 67.

  • Line 350 “wfikkn1’s role” is a bit hard for the reader “wfikkn1 protease inhibitor role” might be clearer.

 We replaced “wfikkn1’s role” with “WFIKKN1 protease inhibitor role”.

Moving on the perspective issue, the current manuscript presents the same weakness that most of the proteomic papers, in the sense that it is very descriptive. The key issue in proteomics is not much the analysis depth, but the biological predictive value of the changes observed in a proteomic screen (and in any omic screen indeed). This seems to be quite an unsaid taboo in omics in general and in proteomics in particular, as I know of only one paper that tackles this issue frontally (doi:10.1016/j.jprot.2021.104178), with quite mixed results indeed. It would therefore be great if the authors could highlight the few original research papers cited in their review (if any) that did perform any kind of validation studies to confirm the proteomic results. I understand that this may be more difficult for people working on animal models, but there is little excuse for the people working on cell lines such as HepG2.If there are no papers with validation experiments, then it must be mentioned.

We agree with reviewer 2 that the final aim of a study that involves proteomics is to highlight new functions or mechanisms of action of proteins. However, these studies (proteomics, functional validation in vitro, validation in vivo…) can be carried out in several stages. Proteomics, an approach without a priori, makes it possible to identify, or to highlight proteins which we would not have hypothesized to be involved. The functional validation of these can be done in a second time. The purpose of this systematic review is to provide for researchers an exhaustive review of the results obtained by proteomic approaches (and possibly reproduced by other groups by these same approaches) and not to describe all the mechanism involved in POP cellular and molecular toxicities. Moreover, if the functional validation of a protein is published in another article, it makes it difficult to rely the functional study to the princeps study based-on proteomic results. In this review, we have specified when results have been confirmed using other biochemical approaches (ex: lines 194-196, 285-286, 406-409, 449-450, 565-567…).

In my opinion, listing the number of proteomic studies whose identified and quantified proteins were then subjected to functional analyses, and discussing and criticizing these results is an excellent idea and deserves to be the subject of a full article. It should be noted that this does not only concern toxicology studies but all proteomic studies, whatever the themes.

In the same trend the authors complain about the wide use of 2D DIGE in the literature that they surveyed and of its lack of depth (which is true) but forget to mention the advantages of this type of approach especially in terms of PTMs. The use of 2D-based proteomics compared to shotgun has been reviewed in many papers. I can propose two of them (doi: 10.1016/j.jprot.2014.03.035 and doi:10.3390/proteomes8030017), but the authors may choose among the many others present in the literature.

We totally agree with reviewer comment.  The aim was not to denigrate 2D-DIGE approach, which may be the approach of choice depending on the problematic studied. Our criticism is based on the fact that the vast majority of teams have used the 2D DIGE approach to study total cell proteomes. The advantages of this approach compared to the others have been added in the text:

However, because of its robustness, its ability to separate proteoforms, and its easy interface with many powerful biochemistry techniques, 2D Gel Electrophoresis could be an approach of choice if used for the appropriate problematic. This concerns least complex proteomes such as secretome or interactome, or even analysis of PTMs (Marcus, K.; Lelong, C.; Rabilloud, T. What Room for Two-Dimensional Gel-Based Proteomics in a Shotgun Proteomics World? Proteomes 2020, 8, 17. https://doi.org/10.3390/proteomes8030017). In the future, it will be necessary for toxicologists to initiate collaboration with proteomic platforms equipped with latest-generation mass spectrometers and bio-informatic tools but also to use the most appropriate technique to answer the biological question asked.

To end with this type of issues, I strongly disagree with the authors regarding their final conclusion (lines 712ff), and I will explain why. Because of the multilayered regulations present in living cells, omic strategies provide molecular evidences, but only biological hypotheses. Consequently, piling omics (as the authors suggest) just piles hypotheses, but does not provide biological evidences. It may just reduce the uncertainty in the translation from molecular events to biological consequences. This problem is especially true for transcriptomics and shotgun proteomics, which completely ignore PTMs and even translational regulations for the former (for the authors’ information, no citation needed here, a discussion of this specific point can be found here: doi: 10.3390/proteomes8030023).

So instead of the complaint on the lack of depth and of the ungrounded (in my opinion) hope in multi-omics, which have been read dozens of times, I suggest that the authors orient their discussion toward more unsung challenges, such as the validation issues (critical in my eyes) and the specific scientific questions risen if we want to consider and tackle potential direct protein modifications by POP-derived reactive metabolites (if this is relevant, owing to the poor chemical reactivity of POPs), as described for other types of pollutants (e.g. in doi: 10.1016/j.toxlet.2013.03.031, doi: 10.1021/tx6003166, doi: 10.1021/tx3002675, doi: 10.1093/toxsci/kfs005, doi: 10.1002/pmic.200401278).

Of course, it is obvious that multi-OMICS approaches do not replace functional studies. Our last paragraph is an opening towards the interest of multi-OMICS approaches, or systems biology, compared to the proteomic approach alone. These multi-omics approaches make it possible to better orient the problem or the hypothesis which will allow subsequent functional validation. Concerning the interest to analyses PTMs, a paragraph is present in the discussion. Concerning the potential direct protein modifications by POP or POP derived reactive metabolites, we identified very few articles that described such modifications (contrary to those described for others chemical molecules such as acrylamide or naphthalene), but we added this interesting point of discussion:

 Moreover, a major concern in proteomic studies is the characterisation of PTMs, such as phosphorylation, glycosylation and acetylation [114]. However, our literature review identifies only two studies that aim to determine the modification of phosphoproteome [44,45] and glycoproteome [45] after POP exposure. As the PTMs of a protein can deter-mine its activity state, location, turnover, and interactions with other proteins, it is now necessary to investigate this using global approaches in order to better understand how exposure to POPs could modify cell machinery and function. In the same manner, POPs, such as other chemical compounds (naphthalene, bisphenol…) are molecules that could directly interact with protein in a covalent manner. It was recently showed that furan-containing compounds (FCCs) can covalently interact with proteins in hepatocytes. Indeed, Li et al. identified 171 lysine-based adducted proteins and 145 cysteine-based adducted proteins by the reactive metabolites of the three FCCs (Li W, Hu Z, Sun C, Wang Y, Li W, Peng Y, Zheng J. A Metabolic Activation-Based Chemoproteomic Platform to Profile Adducted Proteins Derived from Furan-Containing Compounds. ACS Chem Biol. 2022 Apr 15;17(4):873-882. doi: 10.1021/acschembio.1c00917.). These study highlights that the development of efficient chemoproteomic platform to identify adducted proteins and to predict the toxicity of POPs are a very promising approach, complementary to traditional proteomics.
